# Specific Features of Immune Response in Patients with Different Asthma Endotypes Following Immunization with a Conjugate Pneumococcal Vaccine

**DOI:** 10.3390/vaccines13050459

**Published:** 2025-04-25

**Authors:** Anton M. Kostinov, Anna Yu. Konishcheva, Andrey D. Protasov, Mikhail P. Kostinov, Valentina B. Polishchuk, Alexander V. Zhestkov, Natalia E. Yastrebova, Aristitsa M. Kostinova, Zhanar Sh. Musagulova, Ekaterina V. Prutskova

**Affiliations:** 1Allergy Diagnostics Laboratory, Federal State Budgetary Scientific Institution «I. Mechnikov Research Institute of Vaccines and Sera», 105064 Moscow, Russia; ankon81@list.ru; 2Department of Microbiology, Immunology and Allergology, Federal State Budgetary Educational Institution of Higher Education «Samara State Medical University», 443099 Samara, Russia; crosss82@mail.ru; 3Laboratory of Vaccine Prevention and Immunotherapy of Allergic Diseases, Federal State Budgetary Scientific Institution «I. Mechnikov Research Institute of Vaccines and Sera», 105064 Moscow, Russia; monolit.96@mail.ru (M.P.K.); polischook@mail.ru (V.B.P.); 4Department of Epidemiology and Modern Vaccination Technologies, Federal State Autonomous Educational Institution of Higher Education I.M. Sechenov First Moscow State Medical University (Sechenov University), 119991 Moscow, Russia; aristica_kostino@mail.ru; 5Department of Clinical Medicine, Medical University «Reaviz», 443030 Samara, Russia; avzhestkov2015@yandex.ru; 6Laboratory of Immunochemical Diagnostics, Federal State Budgetary Scientific Institution «I. Mechnikov Research Institute of Vaccines and Sera», 105064 Moscow, Russia; yastreb03@rambler.ru; 7Almaty City Health Department, State Municipal Enterprise Under the Right of Economic Management «Children’s City Clinical Infectious Diseases Hospital», Almaty 050026, Kazakhstan; shakizadakizi@mail.ru; 8Department of Pediatrics, Institute of Medicine, Ecology and Physical Education, Federal State Budgetary Educational Institution of Higher Education «Ulyanovsk State University», 432007 Ulyanovsk, Russia; katerina_44@list.ru

**Keywords:** asthma, pneumococcal vaccine, IFN-γ, cytokines, asthma endotypes, atopy, IgE

## Abstract

**Background**: Asthma is a heterogeneous disease characterized by variable bronchial obstruction, hyper-responsiveness, and inflammation. Evaluating the immunological changes following pneumococcal immunization in patients with different asthma endotypes is of great importance. This study aimed to evaluate the effects of PCV13 on the clinical parameters and the changes over time in the levels of the main cytokines in asthma patients. **Methods**: This was a single-center, open-label, non-randomized, prospective, cohort, controlled study of 31 patients aged 18 to 80 with a known diagnosis of asthma. The study subjects were given one injection of PCV13. Their clinical parameters and serum concentrations of certain Th1/Th2/Treg cytokines were assessed over a year following the vaccination. **Results**: Compared to the pre-vaccination period, there was an 81.5% reduction in the number of patients with asthma exacerbations (*p* < 0.001), a 76.5% increase in the number of patients free from hospitalization (*p* < 0.001), and an improvement in the level of asthma control. Positive changes were observed both in patients with T2-high and T2-low asthma; however, only those with T2-low asthma showed a significant improvement in the level of asthma control. Significant changes were reported for IFN-γ: its serum concentrations increased six weeks following the vaccination (*p* < 0.05), primarily in patients with T2-high asthma. **Conclusions**: In asthma patients, immunization with PCV13 was clinically effective, irrespective of the asthma endotype. Its clinical effects were accompanied by a reduction in the rates of exacerbations and hospitalizations and an increase in IFN-γ serum levels. This finding suggests that this cytokine plays an important role in restoring immune response in asthma patients.

## 1. Introduction

Asthma is a heterogeneous disease characterized by variable bronchial obstruction, bronchial hyper-responsiveness, and inflammation caused by various triggers [1]. In asthma patients, dysregulation of immune response is explained by an imbalance in lymphocyte subsets (Th1, Th2, Th17, Treg, and NKT) as well as the components of the innate immune system, including mast cells, basophils, neutrophils, eosinophils, and innate lymphoid cells [2,3,4]. These factors are responsible for molecular differences in patients with somewhat similar clinical features and suggest the predominance of different types of specific responses [5,6]. Today, the identification of the molecular and cellular mechanisms specific to particular asthma endotypes and underlying certain phenotypes of the disease has become a priority because this approach enables a more accurate assessment of a patient’s condition and tailoring the treatment and contributes to the adoption of personalized medicine. According to the Global Initiative for Asthma (GINA) criteria, the most common asthma phenotypes are allergic asthma, non-allergic asthma, cough variant asthma and cough predominant asthma, late-onset asthma, asthma with persistent airflow limitation, and asthma with obesity [1]. However, the identification of asthma endotypes, which is based on the determination of the inflammation type, is of more importance. The following two major asthma endotypes have been established based on the inflammation type: (1) type 2 inflammation-mediated asthma, or eosinophilic asthma (T2-mediated asthma, or T2-high asthma), which is observed in 50–70% of patients, and (2) non-type 2 inflammation-mediated asthma, or non-eosinophilic asthma (non-T2-mediated asthma, or T2-low asthma) [7,8,9]. Due to its higher prevalence, T2-high asthma is considered more established and investigated. However, the mechanisms underlying T2-low asthma have not been well defined and require further investigation to be better understood.

Several long-term follow-up studies of patients immunized against pneumococcal infection demonstrated that pneumococcal vaccines significantly reduced the exacerbation and hospital admission rates in asthma patients in the short and long term [10,11,12,13]. Pneumococcal conjugate and polysaccharide vaccines have also been studied as immunoregulatory therapeutic agents, which contributed to the improvement of asthma clinical features by inducing adaptive immunity to *S. pneumoniae* through modulation of Th1, Th2, Th17, and Treg immune responses [14,15,16,17]. The results of these studies are, however, hard to interpret due to the heterogeneity of asthma, which resulted in the heterogeneity of the study subgroups, and the use of various medicinal products (several types of polysaccharide and conjugate vaccines). 

This explains the non-inclusion of recommendations for the routine pneumococcal vaccination of asthma patients in the current GINA consensus guidelines, which is justified by a lack of relevant large high-quality studies [1].

Considering the above, the objective of this study was to compare the changes over time in the concentrations of the main cytokines responsible for the predominant pattern of inflammation in asthma patients immunized with a conjugate pneumococcal vaccine.

## 2. Materials and Methods

### 2.1. Study Design

A single-center, open-label, non-randomized, prospective, cohort, controlled study was conducted between February 2022 and October 2023.

The inclusion criteria were as follows: ages 18 to 80 years; a known diagnosis of asthma made according to the GINA guidelines; no prior immunization against pneumococcal infection; the absence of other chronic illnesses; the absence of any acute infections at the start of the study; and patients’ written informed consent for participation in the study. 

The non-inclusion criteria were as follows: prior immunization against pneumococcal infection; the use of immunoglobulins or receiving a blood transfusion within the last three months prior to the start of the study; and immunization against flu or other infections covering the study period. 

The exclusion criteria were as follows: violations of the study protocol; the administration of systemic glucocorticoids for a period of 4 weeks following the administration of pneumococcal vaccine; and a patient’s decision to withdraw consent.

Table 1 shows the characteristics of the study subjects.

### 2.2. Study Conditions

The study was conducted at a laboratory of allergy diagnostics using certified equipment provided by the Research Equipment Sharing Center of the Federal State Budgetary Scientific Institution I.I. Mechnikov Research Institute of Vaccines and Sera (Moscow, Russia) and at the Clinical Division of Pulmonology and Allergology, Federal State Budgetary Educational Institution of Higher Education, Samara State Medical University, Ministry of Health of Russia (Samara, Russia). 

### 2.3. Medical Intervention

Overall, 31 asthma patients were included in the study after being discharged from the Clinical Division of Pulmonology and Allergology, Samara State Medical University. The study subjects were immunized against pneumococcal infection with Prevenar 13 (manufactured by NPO Petrovax Pharm LLC, Moscow, Russia), a 13-valent conjugate pneumococcal vaccine (PCV13), which was administered as a single intramuscular injection in accordance with the prescription information [18,19,20].

The clinical picture of the disease was assessed using the following parameters: the number of patients with exacerbations; the mean number of asthma exacerbations per patient over the year prior to the study; the number of patients hospitalized for asthma exacerbations; the mean number of hospitalizations per patient over the year prior to the study; the level of asthma control as assessed by the Asthma Control Questionnaire (ACQ-5); daily doses of ICS; short-acting beta-2 agonists (SABA) and systemic corticosteroids (SCS) administration; and forced expiratory volume in 1 s (FEV_1_).

An exacerbation of asthma was defined as an episode characterized by an appearance or increase in symptoms of rale and wheezing or chest tightness that prompted patients to seek medical help and required treatment modification, which was documented in the subjects’ source documents. A hospitalization was defined as the admission of a study subject to the hospital in order to receive inpatient treatment for asthma, which was documented in the patient’s discharge summary or their outpatient medical chart.

The main study tests and assessments were performed using a set of serum samples obtained from the study subjects and a number of questionnaires evaluating the clinical course of the disease. The serum samples and data from the subjects’ medical charts were analyzed at the following time points of data collection: at baseline, 6 weeks, and 6 and 12 months after the administration of the conjugate pneumococcal vaccine.

During each study visit, blood samples were collected and patient information was obtained to be later included in specially designed questionnaires.

Patients were categorized into T2-high and T2-low asthma endotypes based on a comprehensive evaluation of clinical and biological parameters, as outlined in the methodology of other studies in which the disease was endotyped [7,21,22]. The parameters incorporated encompassed the clinical history (age of onset, atopic status, smoker status, body mass index [BMI], response to inhaled corticosteroid [ICS] therapy, and the level of asthma control), conventional biomarkers (blood eosinophils, serum specific and total IgE levels, forced expiratory volume in 1 s [FEV_1_], forced vital capacity [FVC], and serum levels of individual specific cytokines [for example, IL-4 and IL-6]). This multimodal approach enabled the identification of inflammatory patterns consistent with T2-driven or non-T2-driven asthma in each individual patient, providing a basis for subsequent endotype-specific analysis.

### 2.4. Key Study Outcomes and Outcome Documentation

The serum samples were used to measure the levels of Th1/Th2/Treg cytokines by ELISA (IFN-γ, IL-4, IL-6, IL-8, IL-10, IL-18, tumor necrosis factor α [TNF-α], and monocyte chemoattractant protein 1 [MCP-1]) using suitable commercially available EIA-BEST kits (Vector-Best JSC, Moscow, Russia) in accordance with the package insert instructions. 

Total immunoglobulin (Ig) E levels were measured by ELISA using the suitable commercially available kits (KHEMA LLC, Moscow, Russia) in accordance with the package insert instructions. Elevated IgE levels were defined as serum IgE levels above 130 IU/L (or 312 ng/mL).

The laboratory tests were performed at the I.I. Mechnikov Research Institute of Vaccines and Sera.

### 2.5. Ethical Review

This study was approved by the Local Ethics Council at the I.I. Mechnikov Research Institute of Vaccines and Sera on 18 May 2023 (protocol #8).

### 2.6. Statistical Analysis

Statistical analysis of the study data was performed using Statistica v. 13.1 software (StatSoft Inc., Tulsa, OK, USA). The normality of the quantitative data was assessed using the Shapiro–Wilk test. Normally distributed independent samples were compared using the Student’s *t*-test and the Levene test, which were applied to assess the equality of variances, as well as an analysis of variance (ANOVA). Paired comparisons of non-normally distributed samples were conducted using the Mann–Whitney U test, and three and more non-normally distributed samples were compared using the Kruskal–Wallis test. The dependent samples were analyzed using the paired Student’s *t*-test or the Wilcoxon test. Multiple analyses of the dependent samples were performed using the Friedman test.

## 3. Results

The analysis of the clinical course of asthma in the study subjects revealed the following positive effects of the vaccination with PCV13: an 81.5% reduction in the number of patients with asthma exacerbations (*p* < 0.001) and a 76.5% increase in the number of patients free from hospitalization (*p* < 0.001) over the 1-year follow-up period compared to the 12 months prior to the vaccination (Table 2). 

Additionally, at the end of the follow-up period there was a 44.4% reduction in the mean ACQ-5 score (*p* < 0.001) and a corresponding 50.0% increase in the number of patients with controlled asthma compared to the pre-vaccination values. Improvements in asthma control were associated with changes in related parameters, most notably a decrease in the daily ICS dose and an increase FEV_1_ compared with the pre-vaccination period. The findings demonstrated a 64.3% increase (*p* < 0.05) in the number of patients with low daily ICS doses (or free of ICS), coinciding with a decline in the proportion of patients consuming medium doses. A decline in the daily dosage of ICS was observed within one year of vaccination for 48.4% of patients in the study sample. In a similar vein, a rise in FEV_1_ values was observed in patients following a one-year period of observation in comparison to the pre-vaccination state, with an average increase of 7.7% (*p* < 0.01).

A time-course analysis of the whole cytokine panel showed significant changes primarily in the levels of IFN-γ, a pro-inflammatory Th1 cytokine (Table 3). The analysis of the whole study sample revealed an increase in the median level of this cytokine up to 0.15 [0.00; 2.62] pg/mL at 6 weeks following the vaccination, while at 6 months post-vaccination no significant differences were observed compared to the pre-vaccination levels.

The analysis of the changes in the serum IFN-γ concentrations performed after clustering the patients according to their clinical features and asthma endotypes revealed certain trends. At 6 weeks post-vaccination, the group of patients with T2-high asthma showed an increase in the median IFN-γ serum concentration up to 1.42 [0.00; 2.87] pg/mL (Figure 1). From that moment on until the end of the 1-year follow-up period, the IFN-γ concentrations decreased, and a year after vaccination they reached the pre-vaccination level. It was not so for the patients with T2-low asthma, who did not show this trend.

A considerable number of publications have reported an association of the T2-high asthma endotype with a number of clinical features, particularly high serum levels of total IgE, sensitization to aeroallergens allergens, elevated blood and sputum eosinophil levels, as well as the presence of concomitant diseases, such as atopic dermatitis and allergic rhinitis. Therefore, we assessed these parameters in our study population (Table 4) [23]. After clustering the patients according to their personal history of atopic disorders and total serum IgE levels, the changes in IFN-γ concentrations were shown to be statistically significant. In the patients with a history of atopic disorders and elevated total IgE levels, serum IFN-γ concentrations peaked at week 6 post-vaccination.

The analysis of the correlation between the changes in serum IFN-γ concentrations and the changes in the clinical course of asthma revealed that at week 6 post-vaccination the most significant increase in the serum IFN-γ concentrations was observed in the patients with mild asthma (*p* < 0.05) and those with fewer asthma exacerbations (*p* < 0.01) compared to the pre-vaccination period (Table 5). As early as at month 6 post-vaccination, their serum IFN-γ concentrations did not differ from the pre-vaccination levels. Of note, in the subgroup of patients with a decreased number of hospitalizations, the serum IFN-γ concentrations did not increase to the same extent as in the patients with fewer exacerbations.

The other cytokines evaluated in the study did not show any significant changes during the follow-up period; however, their levels were different in the study subgroups. Thus, the serum concentrations of regulatory IL-10 were significantly higher in patients with increased baseline levels of total IgE (measured at visit 1), and this difference was particularly apparent at month 6 post-vaccination (*p* < 0.05), while no similar correlation was observed between the serum IL-10 concentrations and asthma endotypes (Figure 2). No significant changes were observed in the serum IL-10 concentrations alongside other indicators, including temporal dynamics at any study time point (Table 6).

## 4. Discussion

In chronic bronchopulmonary disorders, such as asthma, *S. pneumoniae* is considered one of the main etiological factors, which further justifies the practical importance of preventive measures in this patient population. Several products are currently employed for pneumococcal vaccination on a global scale, for which a substantial research knowledge base has been amassed: 23-valent polysaccharide pneumococcal vaccine (PPV23) and PCV13. PCV15 and PCV20, which were also developed, were approved by the FDA for use in adults in December 2021, and PCV21 was only approved in June 2024, so they have not yet been widely distributed. Concurrently, researchers have accumulated a substantial evidence base on the effectiveness of pneumococcal vaccines in patients with asthma [24,25]. This vaccination may fulfil several objectives, including the prevention of infectious disease and the reduction in the spread of antibiotic-resistant strains of pneumococcus. Although PPV23 offers broader coverage of pneumococcal serotypes, it does not induce sustained long-term protection. The most effective use of PPV23 is in combination with PCV13, which induces a T-dependent immune response, resulting in the formation of memory B cells and thus longer-term immunity to antigens [26,27].

Despite the different mechanisms of action of these vaccines, the immunization effect on the clinical picture of asthma in patients is consistently positive, regardless of the type of drug administered and the scheme used, but with a clear advantage of sequential administration of PCV13 and then PPV23 [10,11]. In accordance with the recommendations issued by the US Centers for Disease Control and Prevention (CDC), it is advised that adults with asthma should be immunized using either a combination scheme of PCV15, followed at least one year later by PPV23, or a single administration of PCV20 or PCV21. Our findings were consistent with those reported by other authors who had shown that immunization with PCV13 significantly reduced the risks of asthma exacerbations and asthma-related hospitalization [10,11,12,28]. In asthma patients, the immune dysregulation at both the cellular and molecular levels caused by an imbalance in subsets of effector T-cells may promote the suboptimal humoral immune function, especially T-cell-independent immunity against pneumococcal polysaccharides [29]. This is the reason why patients with asthma and atopic diseases might have lower levels of serotype-specific IgG antibodies against vaccine pneumococcal polysaccharide antigens compared to those seen in healthy individuals [30]. Moreover, it was later demonstrated that specific types of inflammation in asthma could, in some way or another, influence patients’ immune response to pneumococcus. Thus, the predominant Th2 cytokine profile, which is characteristic of T2-high asthma, adversely affects the humoral response to pneumococcal polysaccharide antigens, while the normalized production of Th1 cytokines can contribute to the optimal course of this type of immune response [31]. 

In our study, the patients with different asthma endotypes showed similar results regarding a reduction in the number of hospitalizations and asthma exacerbations; however, the patients with T2-low asthma showed a significant improvement in the level of asthma control, as well as an indirectly associated parameter, the FEV_1_ score. This might be partly explained by slightly poorer baseline levels of asthma control, for example lower average ACQ-5 and FEV_1_ scores, in the study subjects with non-eosinophilic asthma. This assumption may be partly supported by the fact that, according to the data on the daily dose of ICS taken before and after vaccination, significant differences in the number of patients with low-dose (or free of) ICS were characteristic only of patients with the T2-high asthma. Furthermore, patients exhibiting a T2-low endotype may, in accordance with the principles of clustering and the definition of endotype per se, be more prone to exhibiting an uncontrolled level of asthma with all its concomitant characteristics [32]. It is therefore hypothesized that a significant improvement in the clinical course of asthma in such patients and their attainment of similar quantitative characteristics to the T2-high endotype may further demonstrate the beneficial effects of pneumococcal vaccination.

It is also worthy of note that the decrease in the number of patients with exacerbations after vaccination in our study led to a reduction in SABA administration in the post-vaccination follow-up period. However, the number of cases of these drugs after vaccination has been found to correlate and correspond to the number of patients with exacerbations that require SABA administration. For example, in our study group there were 27 patients with asthma exacerbations requiring SABA, whereas one year after vaccination the number of such patients decreased to 5, i.e., by 81.5% (*p* < 0.01). This effect was observed in patients with both T2-high and T2-low asthma endotypes. In the post-vaccinal period, the administration of CSC was not a requirement for exacerbation control. Prior to vaccination, cases were sporadic, which precludes objective evaluation of the characteristic changes in this index.

Nevertheless, the improvement in clinical features in asthma patients could be seen as a direct clinical outcome of vaccination, i.e., the prevention of pneumococcal infection and elimination of *S. pneumoniae* from the upper airways [33,34], but could also be attributed to the ability of the study vaccine, particularly its complex of *S. pneumoniae*-specific antigens, to modulate Th1/Th2/Treg immune response by promoting the induction of Treg cells and suppressing Th2 immune response [15,35,36].

Therefore, the amelioration of patients’ conditions following the immunization against *S. pneumoniae* could be a consequence of a potential improvement in the immune function, which, in asthma patients, is affected by dysregulation of immune response due to an imbalance in subsets of effector T-cells (including Th1, Th2, Th17, and NKT-cells) and molecules secreted by them. Thus, our results regarding the changes in serum IFN-γ levels, especially in patients with T2-high asthma, could reflect this phenomenon. Indeed, some authors speculate that that low levels of IFN-γ may promote allergic airway disease by increasing antigen presentation and inflammatory cell recruitment while higher levels of IFN-γ suppress Th2 responses and corresponding Th2 cytokine effects [37]. However, considering the pleiotropic activities of cytokines, the effects of IFN-γ on the pulmonary allergic response remain to be evaluated [38]. 

It should also be noted that the observed transient increase in serum IFN-γ concentration 6 weeks after pneumococcal vaccination is likely to reflect the peak of the early Th1-type cellular immune response, similar to that observed with other conjugate and adjuvanted vaccines. Analogous patterns have been reported in recipients of conjugate vaccines against *Pseudomonas aeruginosa* and synthetic polypeptide influenza vaccines. In such cases, a substantial increase in IFN-γ production has been observed within weeks of vaccination [39,40]. This transient cytokine surge likely initiates longer-lasting immunological changes, including enhanced mucosal immunity, a reduction in bacterial colonization, and the modulation of airway inflammation, which contribute to the sustained clinical improvements observed over the following year [34]. Higher levels of IFN-γ could reduce the rates of viral infections and infections caused by intracellular pathogens, which also lowers the probability of asthma exacerbations and asthma-related hospitalization. Notably, in our study, the characteristic increase in IFN-γ concentrations was primarily observed in patients who had a reduced number of asthma exacerbations compared to the pre-vaccination period.

Our findings regarding the changes in regulatory IL-10 levels do not allow any firm conclusions about its potential role in regulating immune response. It should be noted, however, that IL-10 suppresses hyper-responsiveness and eosinophilia in the airways mainly by inhibiting the antigen-induced recruitment of inflammatory cells. However, some reports showed that elevated levels of IL-10 inhibit Treg cells, which in turn suppress pro-inflammatory signaling pathways that block Th2 cells, macrophages, dendritic cells, and B-cells [41]. At the same time, Treg cells are one of the key players in protection against pneumococcal infection.

## 5. Conclusions

A conjugate pneumococcal vaccine showed high clinical success rates in asthma patients; therefore, it should be considered as a universal option for the prevention of pneumococcal infection, regardless of asthma endotype. Clustering patients by endotypes of asthma allowed the identification of specific patterns and differences in the development of vaccine-induced immunity, which may substantiate the necessity for a more comprehensive investigation of the effectiveness of pneumococcal vaccines taking into account the nature of inflammation in asthma patients, as well as the design and implementation of preventive measures, particularly in the context of the emergence of new variants of conjugated pneumococcal vaccines. The most noticeable finding was the changes in serum IFN-γ concentrations: a significant increase in this cytokine following the administration of PCV13 was primarily observed in patients with T2-high asthma and in those who had fewer asthma exacerbations compared to the pre-vaccination period. This finding might suggest an important role of this cytokine in restoring immune response in asthma patients and possibly indicates the existence of an additional mechanism underlying the clinical benefits of pneumococcal vaccination in asthma. Furthermore, the results obtained provide a rationale for conducting additional studies with the objective of evaluating the efficacy of both combined pneumococcal vaccination schemes and monovaccination with new variants of conjugate vaccines in patients with different asthma endotypes.

## Figures and Tables

**Figure 1 vaccines-13-00459-f001:**
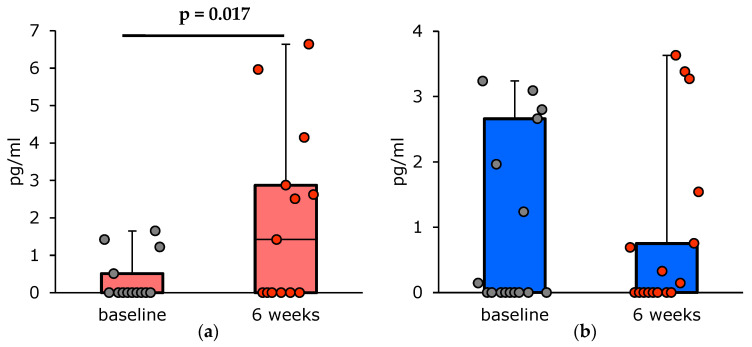
Changes in the serum IFN-γ concentrations (pg/mL) in asthma patients 6 weeks after PCV13 vaccination: (**a**) patients with T2-high asthma (n = 14); (**b**) patients with T2-low asthma (n = 17).

**Figure 2 vaccines-13-00459-f002:**
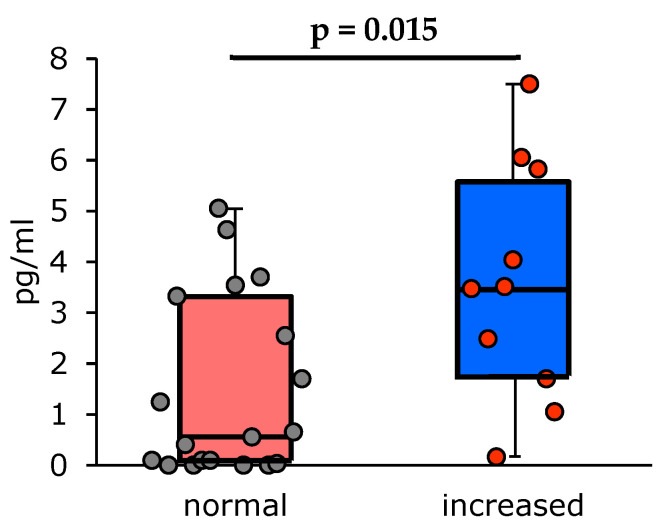
Changes in the serum IL-10 concentrations (pg/mL) in asthma patients 6 months after PCV13 vaccination: patients with normal baseline levels of total IgE (n = 20); patients with increased baseline levels of total IgE (n = 11).

**Table 1 vaccines-13-00459-t001:** Characteristics of the asthma patients included in the study.

Parameters	Values
Number of patients, abs.	31
Asthma duration, years; median (Q1–Q3)	7.0 (3–20)
Number of exacerbations per patient in the last 12 months; mean (standard deviation)	2.3 (2.3)
Number of hospitalizations per patient in the last 12 months; mean (standard deviation)	0.7 (1.1)
Asthma severity; abs. (%):	
mild (intermittent/persistent)	12 (38.7)
moderate	13 (41.9)
severe	6 (19.4)
Level of asthma control; abs. (%):	
controlled	10 (32.2)
partially controlled	2 (6.5)
uncontrolled	19 (61.3)
Personal history of atopic diseases; abs. (%)	10 (32.3)
Asthma endotype; abs (%):	
T2-high asthma	14 (45.2)
T2-low asthma	17 (54.8)
Daily dose of ICS; abs (%):	
none	5 (16.1)
low	9 (29.0)
medium	17 (54.8)
high	0 (0.0)

**Table 2 vaccines-13-00459-t002:** Changes in the asthma clinical features in patients vaccinated with PCV13 (n = 31).

Parameters		Time Periods		Changes
		12 Months Before Vaccination	12 Months After Vaccination	
Number of patients with exacerbations; abs. (%)	T2-high asthma	12 (85.7)	3 (21.4)	↓ (75.0) *p* < 0.001
	T2-low asthma	15 (88.2)	2 (11.8)	↓ (86.7) *p* < 0.001
	Total	27 (87.1)	5 (16.1)	↓ 22 ↓ (81.5) *p* < 0.001
Number of patients free from hospitalization; abs. (%)	T2-high asthma	9 (64.3)	14 (100.0)	↑ (55.6) *p* < 0.001
	T2-low asthma	8 (47.1)	16 (94.1)	↑ (100.0) *p* < 0.001
	Total	17 (54.8)	30 (96.8)	↑ 13 ↑ (76.5) *p* < 0.001
Number of patients with controlled asthma; abs. (%)	T2-high asthma	8 (57.1)	7 (50.0)	↓ (12.5)
	T2-low asthma	2 (11.8)	8 (47.1)	↑ (300.0) *p* < 0.05
	Total	10 (32.3)	15 (48.4)	↑ 5 ↑ (50.0)
Number of patients on low daily doses of ICS (or free of ICS) at the end of the follow-up period; abs. (%)	T2-high asthma	5 (35.7)	12 (85.7)	↑ 7 ↑ (140.0) *p* < 0.05
	T2-low asthma	9 (52.9)	11 (64.7)	↑ 2 ↑ (22.2)
	Total	14 (45.2)	23 (74.2)	↑ 9 ↑ (64.3) *p* < 0.05
Mean FEV_1_ at the end of the follow-up period; (SD) [min; median; max]	T2-high asthma	73.41 (15.38) [38.10; 79.50; 88.90]	77.64 (14.10) [57.00; 79.55; 96.20]	↑ (5.8)
	T2-low asthma	70.51 (17.17) [31.60; 74.00; 92.80]	77.12 (15.87) [40.90; 83.90; 95.60]	↑ (9.4) *p* < 0.01
	Total	71.82 (16.19) [31.60; 74.90; 92.80]	77.35 (14.85) [40.90; 82.70; 96.20]	↑ (7.7) *p* < 0.01
Mean ACQ-5 score at the end of the follow-up period; (SD) [min; median; max]	T2-high asthma	1.10 (1.06) [0.00; 0.60; 3.20]	0.73 (0.52) [0.00; 0.70; 1.80]	↓ (33.6)
	T2-low asthma	2.18 (1.12) [0.00; 2.00; 4.60]	1.12 (1.20) [0.00; 0.80; 4.20]	↓ (48.6) *p* < 0.01
	Total	1.69 (1.21) [0.00; 1.80; 4.60]	0.94 (0.96) [0.00; 0.80; 4.20]	↓ (44.4) *p* < 0.01

↑ — the designation of a change in the setpoint when comparing two time periods, which indicates its increase; ↓ —the designation of a change in the setpoint when comparing two time periods, which indicates its decrease.

**Table 3 vaccines-13-00459-t003:** Serum concentrations of cytokines (pg/mL) in asthma patients vaccinated with PCV13 (n = 31).

Cytokines	Time Points			
	Baseline	6 Weeks	6 Months	12 Months
IL-4	1.33 [0.00; 2.87]	1.18 [0.50; 2.26]	1.56 [0.09; 2.53]	0.78 [0.00; 1.87]
IL-6	1.44 [0.82; 2.62]	2.05 [0.88; 3.52]	1.52 [0.80; 2.82]	1.33 [0.47; 2.25]
IL-8	5.93 [4.84; 8.67]	6.63 [5.19; 9.20]	6.87 [4.90; 8.37]	6.01 [2.69; 7.94]
IL-10	1.46 [0.51; 2.68]	1.08 [0.16; 3.38]	1.70 [0.09; 3.54]	0.57 [0.00; 2.57]
IL-18	155.50 [99.30; 248.00]	210.72 (124.35) [0.00; 178.50; 525.00]	182.06 (108.63) [4.63; 167.00; 461.00]	135.00 [87.40; 216.00]
IFN-γ	0.00 [0.00; 1.24]	0.15 * [0.00; 2.62]	0.00 [0.00; 1.15]	0.00 [0.00; 0.97]
TNF-α	5.23 [3.61; 6.94]	5.50 [3.09; 7.11]	5.48 [5.04; 6.91]	6.02 [5.04; 7.48]
MCP-1	144.00 [84.10; 185.00]	167.39 (89.52) [4.40; 180.00; 361.00]	161.50 [71.40; 193.00]	162.62 (132.87) [0.00; 160.00; 519.00]

Normally distributed data are presented as mean (SD) [min; median; max], and non-normal data are presented as median [Q1; Q3]; * *p* < 0.05 is the significance level for the differences between the values obtained at specified time points and others.

**Table 4 vaccines-13-00459-t004:** Changes in the serum IFN-γ concentrations (pg/mL) in asthma patients with different asthma endotypes and other disease-related parameters following PCV13 vaccination (n = 31).

Parameters	Time Points				
	Baseline	6 Weeks	6 Months	12 Months	*p* Value
Asthma endotype					
T2-high asthma	0.00 [0.00; 0.51]	1.42 [0.00; 2.87]	0.00 [0.00; 0.00]	0.00 [0.00; 0.51]	**0.003**
T2-low asthma	0.00 [0.00; 2.66]	0.00 [0.00; 0.75]	0.00 [0.00; 2.41]	0.00 [0.00; 2.33]	0.480
History of atopic disorders					
No	0.00 [0.00; 1.60]	0.00 [0.00; 1.15]	0.00 [0.00; 2.41]	0.00 [0.00; 1.20]	0.589
Yes	0.00 [0.00; 1.22]	2.52 (2.40) [0.00; 2.51; 6.64]	0.00 [0.00; 0.00]	0.00 [0.00; 0.69]	**0.002**
Level of total IgE at visit 1 (baseline)					
Normal	0.00 [0.00; 1.24]	0.24 [0.00; 2.51]	0.00 [0.00; 2.30]	0.00 [0.00; 1.26]	0.154
Increased	0.00 [0.00; 1.42]	0.00 [0.00; 4.15]	0.00 [0.00; 0.00]	0.00 [0.00; 0.51]	**0.025**

Normally distributed data are presented as mean (SD) [min; median; max], and non-normal data are presented as median [Q1; Q3]; bold highlighting—statistically significant differences.

**Table 5 vaccines-13-00459-t005:** Changes in the serum IFN-γ concentrations (pg/mL) in asthma patients with different clinical features of asthma following PCV13 vaccination (n = 31).

Parameters	Time Points				
	Baseline	6 Weeks	6 Months	12 Months	*p* Value
Asthma severity					
mild	0.00 [0.00; 1.23]	0.78 [0.00; 3.51]	0.00 [0.00; 2.30]	0.00 [0.00; 1.79]	**0.011**
moderate	0.00 [0.00; 1.96]	0.00 [0.00; 0.75]	0.00 [0.00; 2.69]	0.00 [0.00; 1.14]	0.963
severe	0.00 [0.00; 0.15]	1.56 (1.49) [0.00; 1.60; 3.27]	0.00 [0.00; 0.00]	0.00 [0.00; 0.69]	**0.036**
Changes in the number of exacerbations (compared to the 12 months before the vaccination)					
decreased	0.00 [0.00; 1.42]	0.51 [0.00; 2.87]	0.00 [0.00; 2.30]	0.00 [0.00; 1.26]	**0.006**
unchanged	0.00 [0.00; 0.00]	0.00 [0.00; 0.00]	0.00 [0.00; 0.00]	0.00 [0.00; 0.00]	–
increased	–	–	–	–	–
Changes in the number of hospitalizations (compared to the 12 months before the vaccination)					
decreased	0.00 [0.00; 0.00]	0.15 [0.00; 1.54]	0.00 [0.00; 0.00]	0.00 [0.00; 0.97]	0.432
unchanged	0.07 [0.00; 1.13]	0.16 [0.00; 3.51]	0.00 [0.00; 2.30]	0.00 [0.00; 0.88]	**0.007**
increased	–	–	–	–	–

Normally distributed data are presented as mean (SD) [min; median; max], and non-normal data are presented as median [Q1; Q3]; bold highlighting—statistically significant differences.

**Table 6 vaccines-13-00459-t006:** Changes in the serum IL-10 concentrations (pg/mL) in asthma patients with different asthma endotypes and clinical features following PCV13 vaccination (n = 31).

Parameters	Time Points	
	6 Weeks	6 Months
Asthma endotype		
T2-high asthma	2.12 (2.00) [0.00; 1.61; 5.49]	2.86 (2.61) [0.00; 2.55; 7.60]
T2-low asthma	0.61 [0.00; 2.88]	1.05 [0.03; 3.54]
*p* value	0.430	0.196
History of atopic disorders		
No	1.08 [0.20; 4.05]	1.42 (1.75) [0.00; 0.53; 5.05]
Yes	0.71 [0.00; 1.67]	0.53 [0.09; 2.87]
*p* value	0.391	0.302
Level of total IgE at visit 1 (baseline)		
Normal	0.88 [0.00; 2.88]	0.56 [0.03; 3.32]
Increased	2.27 (2.02) [0.00; 1.67; 5.49]	3.58 (2.35) [0.16; 3.49; 7.50]
*p* value	0.282	**0.015**

Normally distributed data are presented as mean (SD) [min; median; max], and non-normal data are presented as median [Q1; Q3]; bold highlighting—statistically significant differences.

## Data Availability

The data presented in this study are available upon request from the corresponding author.

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
