# Peer review of "Specific Features of Immune Response in Patients with Different Asthma Endotypes Following Immunization with a Conjugate Pneumococcal Vaccine"

_vaccines, 2025, doi:10.3390/vaccines13050459_

Round 1

Reviewer 1 Report

Comments and Suggestions for Authors

1) Were systemic steroids within 4 weeks of PCV13 administration an exclusion if so please state, since it should be.

2) How were asthma endotypes at baseline separated between T2high/T2low. Please expand.

3) What were the subject's Spirometry measurements at baseline and throughout study? Was FeNO and blood eos measured? Was IL-5, IL-13, IL-17 measured? Was the amount of SABA rescue use and/or SCS use for exacerbations reduced after vaccination? Why not report these measurments? Results may be more significant if these other markers were measured.

4) Please expand the similarities and difference between different pneumococcal vaccines (Polysacc vs Conjugate) with pro/con as to their effectiveness. Why or why not same results?

5) Although IFN-g increased, it was only at 6 weeks and not longer in time as the asthma control suggests, why? Why were the moderate/severe asthmatics and/or decreased hospitalizations not improved like IFN-g?

6) What baseline characteristic predicted best response to PCV13 vaccination response with reductions of asthma exacerbations?

Author Response

I would like to express my sincere gratitude for your valuable time and thorough feedback on specific aspects of the manuscript and research. We have carefully considered all your comments and suggestions, and below, you will find a detailed description of each of the points.

Comment 1: Were systemic steroids within 4 weeks of PCV13 administration an exclusion if so please state, since it should be

Response 1: The administration of SCS for a period of four weeks following the administration of PCV13 was designated as an exclusion criterion (starting on line 87). It is interesting to note that, in the patient group under study, the requirement for SCS was very limited, with ICS alone being sufficient

Comment 2: How were asthma endotypes at baseline separated between T2high/T2low. Please expand

Response 2: The Materials and Methods section has been updated to include additional information on the endotyping of the disease (starting on line 125). In particular, reference was made to a number of individual papers that were referenced for this process, namely 10.1007/s12016-018-8712-1. The initial objective was to identify asthma phenotypes as listed by GINA, among other sources. However, the limited sample size of patients precluded the attainment of sufficiently reliable results specific to certain phenotypes. Consequently, the endotyping approach was selected as a more comprehensive and upper-level strategy in our study. This approach is predicated on both clinical characteristics and specific biomarkers. A comprehensive data set was analysed, encompassing primary markers (presence of atopy, eosinophil count, total and specific IgE levels) and secondary markers (ICG intake and doses, lung function (e.g. FEV1), levels of certain cytokines, smoking status, BMI, etc.). The patients were then clustered according to the values and combinations of these markers.

Comment 3: What were the subject's Spirometry measurements at baseline and throughout study? Was FeNO and blood eos measured? Was IL-5, IL-13, IL-17 measured? Was the amount of SABA rescue use and/or SCS use for exacerbations reduced after vaccination? Why not report these measurments? Results may be more significant if these other markers were measured

Response 3: Relevant material has been added to the Results (starting on line 168) and Discussion (starting on line 278) sections. The cytokine pool listed (IL-5, IL-13 and IL-17) was unfortunately not examined, nor was FeNO. Serum eosinophil levels were not determined in the dynamics, while baseline values were also used for asthma endotyping. However, information on pulmonary function tests (e.g. FEV1) as well as the ICS used in the dynamics was added to the manuscript. The use of SCS in our study group is an individual case, so statistical treatment is not possible. Certain conclusions on SABA have also been added.

In general, our aim is to provide new material, bearing in mind that we have already published some statistics on a comprehensive study of this issue, and therefore the range of parameters studied was initially broader than what is directly stated in this manuscript, so as not to overlap with other articles.

Comment 4: Please expand the similarities and difference between different pneumococcal vaccines (Polysacc vs Conjugate) with pro/con as to their effectiveness. Why or why not same results?

Response 4: The differences between the vaccines used, as well as the vaccination schemes against pneumococcal infection in patients with asthma, were described in the Discussion section (starting on line 241)

Comment 5: Although IFN-g increased, it was only at 6 weeks and not longer in time as the asthma control suggests, why? Why were the moderate/severe asthmatics and/or decreased hospitalizations not improved like IFN-g?

Response 5: The relevant material has been added to the Discussion section (starting on line 318). We potentially associate the transient increase in serum IFN-γ observed 6 weeks after pneumococcal vaccination with a peak Th1-type cellular response, as seen with other conjugate and adjuvanted vaccines.This short-term cytokine surge may initiate longer-lasting immune changes—such as enhanced mucosal immunity and reduced airway inflammation—that help sustain clinical improvements over time.

Comment 6: What baseline characteristic predicted best response to PCV13 vaccination response with reductions of asthma exacerbations?

Response 6: In conclusion (starting on line 343), a more detailed phrase was incorporated regarding the effectiveness of the vaccine and the type of inflammation observed in patients. It is imperative to reiterate that the present study did not reveal a statistically significant discrepancy in the clinical effects of vaccination among patients with varying asthma endotypes. Consequently, the conjugate vaccine has been shown to be equally efficacious in patients with both T2 high and T2 low endotypes. This finding is in contrast to the our results observed with the polysaccharide vaccine, which are scheduled to be prepared and published in the near future. In the case of PPV23, no significant differences were found in the reduction in the number of hospitalisations and exacerbations in patients with the endotype of T2 low endotype. This finding prompts questions regarding the efficacy of combined vaccination schemes (PCV13/PPV23 or PCV15/PPV23) in such patients.

Reviewer 2 Report

Comments and Suggestions for Authors

Thank you for this fascinating review of the effect of Conjugate vaccine (PCV 13) on asthma.

You have found that there is a clear benefit in asthma exacerbations for both T2 high and low patients, but quality of life in those with T2 low inflammation.

I also found the discussion about low level interferon levels possibly predicting type 2 inflammation fascinating.

In the discussion, it might be appropriate to discuss that PCV 13 has now mostly been replaced by PCV 20 and 21. Some conjecture about  PPS 23 and literature review on this vaccine in this clinical situation is also indicated.

I recognize the inclusion was no prior pneumococcal vaccination, but in real life, patients get PCV 13 after previous PPS 23..

IN addition, some discussion of future thoughts on research and implications for treatment should also be added.

Author Response

I would like to express my sincere gratitude for your valuable time and thorough feedback on specific aspects of the article and research. We have carefully considered all your comments and suggestions, and below, you will find a detailed description of each of the points.

Comment 1: In the discussion, it might be appropriate to discuss that PCV 13 has now mostly been replaced by PCV 20 and 21. Some conjecture about PPS 23 and literature review on this vaccine in this clinical situation is also indicated

Response 1: Additional material has been added to the Discussion section (starting on line 241) regarding vaccines administered. While it is true that conjugate vaccines have seen significant progress in recent years, the distribution of new preparations is not yet as widespread. Furthermore, combined PCV/PPV vaccination schemes remain a valid option.

Comment 2: I recognize the inclusion was no prior pneumococcal vaccination, but in real life, patients get PCV 13 after previous PPS 23.

Response 2: Some more information has been added to the Discussion section (starting on line 241). You are right: PСV13 should be used with PPV23. This manuscript only presents part of the results from the work we have been doing in recent years on PCV13. The aim of this work is to compare different vaccination schemes against pneumococcal infection in patients with asthma. In our patient sample, there are four groups that received, among others, the combined schemes PCV13/PPV23 and PPV23/PCV13. We are still analysing the data, but we hope to publish new data soon that includes other vaccines and treatment schemes.

According to official statistics on vaccination, in many countries, including Russia, patients in risk groups are given a single vaccination with PCV13 or PPV23, despite all existing recommendations. The combined scheme is used much less frequently, which is somewhat disappointing and also makes its own adjustments in terms of studies.

Comment 3: IN addition, some discussion of future thoughts on research and implications for treatment should also be added.

Response 3: The Conclusion section is supplemented by the potential practical relevance and possible options for future research.

Round 2

Reviewer 1 Report

Comments and Suggestions for Authors

Approved